# The Role of Extracellular Vesicles (EVs) in Chronic Graft vs. Host Disease, and the Potential Function of Placental Cell-Derived EVs as a Therapeutic Tool

**DOI:** 10.3390/ijms24098126

**Published:** 2023-05-01

**Authors:** Mor Zavaro, Ayelet Dangot, Tali Hana Bar-Lev, Odelia Amit, Irit Avivi, Ron Ram, Anat Aharon

**Affiliations:** 1Hematology Research Laboratory, Hematology Division, Tel-Aviv Sourasky Medical Center, Tel Aviv 6423906, Israel; morzabaro@gmail.com (M.Z.);; 2The Sackler Faculty of Medicine, Tel Aviv University, Tel Aviv 6195001, Israel; iritavi@tlvmc.gov.il (I.A.); ronr@tlvmc.gov.il (R.R.); 3The BMT Unit, Tel Aviv Sourasky Medical Center, Tel Aviv 6423906, Israel; 4Hematology Department, Tel Aviv Sourasky Medical Center, Tel Aviv 6423906, Israel

**Keywords:** chronic graft-versus-host diseases (cGVHD), extracellular vesicles (EVs), inflammation, fibrosis, placenta

## Abstract

Chronic graft-versus-host disease (cGVHD) presents with dermal inflammation and fibrosis. We investigated the characteristics of extracellular vesicles (EVs) obtained from cGVHD patients, and their potential effects on human dermal fibroblast (NHDF) cells. The anti-inflammatory and anti-fibrotic effects of placental EVs were also explored given their known anti-inflammatory properties. Fourteen cGVHD patients’ EVs contained higher levels of fibrosis-related proteins, TGFβ and α-smooth muscle actin (αSMA), compared to EVs from thirteen healthy subjects. The exposure of NHDF cells to the patients’ EVs increased the NHDF cells’ TGFβ and αSMA expressions. Placental EVs derived from placental-expanded cells (PLX) (Pluri Inc.) and human villous trophoblast (HVT) cells expressing the mesenchymal markers CD29, CD73, and CD105, penetrated into both the epidermal keratinocytes (HACATs) and NHDF cells. Stimulation of the HACAT cells with cytokine TNFα/INFγ (0.01–0.1 ng/µL) reduced cell proliferation, while the addition of placental EVs attenuated this effect, increasing and normalizing cell proliferation. The treatment of NHDF cells with a combination of TGFβ and placental HVT EVs reduced the stimulatory effects of TGFβ on αSMA production by over 40% (*p* = 0.0286). In summary, EVs from patients with cGVHD can serve as a biomarker for the cGVHD state. Placental EVs may be used to regulate dermal inflammation and fibrosis, warranting further investigation of their therapeutic potential.

## 1. Introduction

An allogeneic hematopoietic cell transplant (HCT) is a vital therapeutic option for a variety of hematologic disorders. Graft-versus-host disease (GVHD) remains the most frequent and serious complication following allogeneic HCT. Chronic GVHD (cGVHD) is a multi-organ, immune-mediated, life-threatening disorder that occurs in 30–70% of patients undergoing HCT [1]. cGVHD is associated with substantial morbidity and overall poor outcomes, and is a major cause of late-onset post-HCT non-relapse mortality. cGVHD progresses from inflammation to fibrosis, and is characterized by aberrant tissue repair promoted by pro-inflammatory cytokines that induce the migration of activated immune cells toward target organs, with subsequent secretion of cytokines [2]. Th17/Tc17 cells thereby produce GM-CSF, IL-22, IL-13, IFN-γ, and TNFα [3], while activated macrophages produce transforming growth factor-β (TGF-β) and platelet-derived growth factor (PDGF), leading to fibroblast activation [4]. Fibroblast activation occurs mainly via the TGF-β signaling of the SMAD proteins that triggers a collagen production cascade [5,6]. Collagen production begins with the formation of α-smooth muscle actin (αSMA), and continues with procollagen, followed by matrix collagen, ultimately causing tissue stiffness. In parallel, SMAD7 is released from the nucleus and down regulates this process by inhibiting the phosphorylation of SMAD2/3 and inducing the degradation of the TGF-β receptor [7]. This pro-fibrotic phase may affect the joints, lungs, or skin, and inflict life-threatening morbidity [1,8,9]. While the first-line therapy for cGVHD is steroids, approximately 50% of patients become either steroid-dependent or steroid-resistant and require further intervention for disease control. Several drugs for treating steroid-refractory GVHD have been recently approved by the FDA, but more efficacious treatment for cGVHD remains an important unmet need [4]. 

Mesenchymal stem cells (MSCs) are multipotent, non-hematopoietic stem cells that can be isolated from multiple tissues and be applied for tissue repair and immunomodulatory responses [10]. Immunosuppressive effects of MSCs include both the inhibition of T cell proliferation and cytotoxic activity [11]. While MSCs have been extensively studied in the prevention and treatment of acute GVHD, as well as being used therapeutically for steroid-refractory acute GVHD, few studies have assessed their efficacy in the setting of cGVHD. For example, the placenta can be used as a source for MSC-like cells (P-MSCs) which can modulate immune cell functions [11,12]. Placental MSCs express the human leukocyte antigen (HLA)-G, a member of the major histocompatibility complex (MHC) class I, but not MHC class II antigens, and display anti-inflammatory and pro-regenerative properties [12]. In the last decade, mesenchymal-like cells obtained from healthy full-term human placentas were used in clinical trials as treatments for multiple sclerosis [13], as well as in mice models of ischemic stroke [14]. 

Extracellular vesicles (EVs) are membrane vesicles found in circulating blood and in other biological fluids under normal physiologic conditions, and their levels increase in a variety of diseases [15]. Both small EVs (<150 nm, include exosomes) and large EVs (150–1000 nm) contain different proteins (e.g., growth factors, cytokines), fragmented DNA, RNA, and miRNA, which regulate cell–cell communication and function [16,17]. In addition to our research, others have shown that EVs can serve as biomarkers of disease severity in various pathologies [18,19]. As such, EVs may also have a therapeutic potential, considering the fact that MSC-derived EVs maintain many of the therapeutic properties of their parent cells [20]. Several studies demonstrated the therapeutic advantage of using placental EVs in animal models of ischemic stroke [14] and myelin regeneration in multiple sclerosis [21], as well as a topical application to aging skin in humans [22]. Because EV characteristics reflect those of their parental cells, their location in the placenta and early pregnancy stage can influence the cell profile and function [23,24].

This is a two-part study. The first part investigates the characteristics of cGVHD patients’ EVs compared to those of heathy controls. The second part analyzes EVs derived from placental cells and characterizes their therapeutic properties. Specifically, the current study explored the therapeutic potential of EVs originating from two distinct sources of placental cells: (1) human term placental-expanded (PLX) mesenchymal-like adherent stromal cells and (2) early pregnancy stage of human villus trophoblast (HVT) cells that line the placental villi and are in direct content with maternal circulation. We assume that placental EVs express the immunosuppressive antigen characteristic of their parental cell surfaces, and may possibly reduce the inflammation and fibrosis related to cGVHD disease.

## 2. Results

### 2.1. Characteristics of cGVHD Patient EVs Compared to EVs of Healthy Controls

#### 2.1.1. cGVHD Patient Characteristics

cGVHD severity was determined according to the NIH Consensus Criteria [25]. EVs were obtained from fourteen patients with severe cGVHD and thirteen age- and gender-matched HCs. Table 1 provides the patients’ clinical characteristics.

#### 2.1.2. Patient EV Characteristics

##### Patient EV Size and Exosome Markers

Nanoparticle tracking analysis (NTA) displayed similar concentrations and sizes of EVs in platelet-poor plasma (PPP) obtained from cGVHD patients and healthy controls (HCs) (Figure 1a,b). The majority of the EVs obtained from both groups were smaller than 150 nm, and only 14% of the EVs were larger than 150 nm (Figure 1c). The EVs of both groups expressed similar levels of the exosome markers CD63 and CD81 (expressed as a ratio of actin) (Figure 1d,e, Appendix A).

##### Fibrosis-Related Proteins Cargo of cGVHD Patient EVs

The EVs derived from patients with cGVHD contained significantly higher levels of fibrosis-related proteins compared to the EVs obtained from the HC (Appendix A). The TGF-β of the HC EVs were low (1.04 ± 1.95), but significantly higher in the EVs of the cGVHD patients (2.65 ± 2.56, *p* = 0.027, Figure 2a). The SMAD7 levels were also low in the HC EVs (0.78 ± 0.63), but they were non-significantly higher in the cGVHD patient EVs (1.52 ± 1.18, *p* = 0.065, Figure 2b). The mean αSMA levels of the cGVHD patient EVs were significantly higher compared to EVs from the HC (2.36 ± 2.11 vs. 0.60 ± 0.59, respectively, *p* = 0.0011, Figure 2c). The TGF-β levels in the EVs of both the HC and the cGVHD patients correlated positively with the SMAD7 levels (Figure 2d) (Spearman r = 0.5315, *p*= 0.0036), but not with the αSMA levels.

#### 2.1.3. Patient EV Effects on Dermal Fibroblasts (NHDF)

Exposure of dermal fibroblasts to EVs obtained from patients with cGVHD, for 48 h, induced statistically significant higher levels of TGF-β (2.76 ± 1.99) compared to that observed following exposure to HC-derived EVs (0.45 ± 0.52, *p* = 0.0068, Figure 2e, Appendix A).

A trend of increasing SMAD7 was detected after exposure of NHDF cells to the patient EVs. Specifically, of the seven patients tested, EVs from four of them induced >3 times the production of SMAD7 (18.75 ± 2.41, *p* = 0.0159) compared to HC EVs (4.78 ± 3.67), while addition of EVs from the other three patients induced a reduction in SMAD7 production compared to the HC EVs (Figure 2f).

A trend of increasing αSMA was detected after the exposure of NHDF cells to the patient EVs (4.74 ± 6.75) and HC EVs (1.56 ± 1.38). However, exposure of NHDF cells to two of the patients’ EVs induced >10 times higher expression of αSMA compared to the other five patient EV samples (14.56 ± 0.8 vs. 0.80 ± 0.71, respectively) or compared to HC EVs (1.56 ± 1.38) (Figure 2g).

### 2.2. The Therapeutic Properties of Placental EVs 

#### 2.2.1. Placental EV Characteristics and Function

##### Placental EV Size and Exosome Markers

The characteristics of placental EVs derived from two sources of placental cells (HVT and PLX) were assessed via transmission electron microscopy (TEM). The images from both placental EV samples representing EVs of varied sizes are shown in Figure 3a,b. NTA displayed similar concentrations and mean EV sizes in samples isolated from PLX cells (187.5 ± 14.14 nm) and HVT cells (179.4 ± 10.80 nm). The majority of EVs from both sources were above 150 nm (Figure 3c–g). Interestingly, EVs obtained from the PPP of cGVHD patients and HC were much smaller than EVs obtained from placental cell cultures (Appendix A). EVs obtained from both placental cell cultures expressed similar levels of the exosome markers CD63 and CD81, (Figure 3h,i, Appendix A).

##### MSC Marker Expression on Placental Cells and Their EVs

The expression of the MSC markers was measured via flow cytometry using specific fluorescent antibodies. The PLX and HVT placental cells and their related EVs expressed the CD73, CD105, CD29, and PDL1 markers, while the EVs demonstrated non-significantly lower levels of CD73 and CD105 expression (Figure 4a,b) compared to their parental cells, as well as significantly reduced expressions of CD29 and PDL1 compared to their parental cells (Figure 4c,d). HLA-DR (negative control, non-mesenchymal marker) expression was high in the HVT cells (30% of labeled cells) compared to its expression in PLX cells, but low in their related EVs (10%, *p* < 0.05, Figure 4e).

#### 2.2.2. Interaction of Placental EVs with Dermal Cells

The ability of placental EVs to bind and penetrate dermal cells was assessed through fluorescence microscopy. The images demonstrated that green placental EVs originating from calcein-AM-labeled placental cells bound to dermal cell membranes. The cell membranes were pre-labeled with DID (red), and the co-localization of the green EVs with red membranes is depicted in yellow and the penetration of the green EVs into the cell nucleus pre-labeled with Hoechst (blue) is depicted in pink staining in Figure 5a. Flow cytometer analysis showed the penetrance of green PLX EVs or green HVT EVs to non-labeled HaCaT cells in 68.30 ± 10.33% and 92.27± 6.130, respectively (Figure 5b, Appendix A). Interestingly, adding the supernatant fluids that were collected after the centrifugation of HVT EVs stained 52.68 ± 22.30% of the HaCaT cells, confirming the presence of small EVs in the supernatant (Figure 5b). An addition of green PLX EVs to non-labeled normal human dermal fibroblast (NHDF) cells stained 37.58 ± 11.49% of the cells. Adding green HVT EVs to these cells stained 72.31 ± 26.06% of the cells, while the supernatants of HVT EVs stained 24.02 ± 10.85% (Figure 5c). Next, we validated the effects of placental EVs on dermal cell functions. Placental EVs obtained from both PLX or HVT cells had no effect on HaCaT cell proliferation and activity. Both PLX and HVT EVs increased more than twice in their percentage levels of the anti-apoptotic genes (BCL2/BAX ratio) compared to non-treated cells (Figure 5d). Evaluation of the anti-fibrotic effect of placental EVs revealed that adding PLX EVs to NHDF cells reduced the expression of TGF-β, while HVT EVs did not significantly affect NHDF cell TGF-β expression. Both types of placental EVs originating from PLX and HVT cells increased the level of the SMAD7 inhibitor protein, but did not affect αSMA protein levels (Figure 5e).

#### 2.2.3. The Effects of Cytokine-Induced Inflammation and Fibrosis on Dermal Cell Models

In order to validate the ability of placental EVs to control the effects of cytokines on dermal cell models, we needed to first define the effects of the related cytokines, such as TNFα, IFNγ, and TGF-β, on dermal inflammation and fibrosis. Dermal keratinocyte (HaCaT) cells were exposed to increasing doses of two specific cytokines, TNFα and IFNγ, which are known to be associated with the inflammatory state in cGVHD. As measured by an XTT assay, exposure to increasing doses of TNFα and IFNγ (0.01 ng/µL, 0.1 ng/µL, 1 ng/µL) for 24 h reduced HaCaT cell proliferation to 70% (*p* = 0.008), 34% (*p* = 0.008), and 45% (*p* = 0.0139), respectively, compared to non-treated cells (Figure 6a). Exposure of keratinocytes for 48 h to increasing doses of TNFα and IFNγ significantly increased the apoptosis of HaCaT cells by 4- to 8-fold compared to untreated cells (0.01 ng/µL, *p* = 0.0286; 0.1 ng/µL, *p* = 0.0286 and 1 ng/µL, *p* = 0.0571), as evidenced by the Annexin/PI assay (Figure 6b, Appendix A). Exposure of keratinocytes to increasing doses of TNFα and IFNγ (0.01 ng/µL, 0.1 ng/µL, 1 ng/µL) for 6 h significantly decreased the ratio between anti- and pro-apoptotic gene expression by 52% (*p* = 0.0112), 43% (*p* = 0.0179), and 48% (*p* = 0.0179), respectively, compared to untreated controls, as measured by RT-PCR (Figure 6c). Exposure of dermal fibroblast (NHDF) cells to 0.25 ng/µL of TGF-β induced a significant increase in αSMA proteins compared to non-treated cells (*p* = 0.0117), and a trend of increment in SMAD7 levels (Figure 6d, Appendix A).

#### 2.2.4. Effects of Placental EVs on Cytokine-Induced Keratinocyte Cell Functions

The addition of PLX EVs to cells stimulated with increasing doses of cytokines (TNFα and INFγ 0.01 ng/µL and 0.1 ng/µL) attenuated the cytokines’ anti-proliferative effects, resulting in an increased proliferation of HaCaT cells. Specifically, while TNFα and INFγ (0.01 ng/µL) reduced proliferation from 100% to 70 ± 17%, the combination of cytokines with PLX EVs increased cell proliferation to 102 ± 10% (*p* = 0.0159). The addition of TNFα and INFγ (0.1 ng/µL) to HaCaT cells reduced their proliferation from 100% to 34 ± 30%, and the combination of cytokines with PLX EVs increased cell proliferation to 87 ± 19% (*p* = 0.0571). The addition of HVT EVs was less effective, with an increasing trend of proliferation having been documented solely for the combination of HVT EVs with 0.1 ng/µL TNFα and INFγ, compared to this dose of cytokines alone. However, placental EVs of both types did not alter the anti-proliferative effects induced by the high dose of TNFα and INFγ (1 ng/µL) (Figure 7a).

Placental EVs did not affect keratinocyte apoptosis in the low and moderate doses of TNFα and INFγ. Cell apoptosis (measured by annexin/PI) was non-significantly reduced (>30%) when PLX EVs or HVT EVs were added to the high cytokine dose (TNFα and INFγ 1 ng/µL) (Figure 7b). However, cytokine-induced HaCaT cells that were exposed to PLX EVs or HVT EVs increased the expression of anti-apoptotic genes. At low doses of cytokines, PLX EVs significantly increased the HaCaT cell anti-apoptotic effect by 42% (*p* = 0.0571) and HVT EVs induced a trend toward an increase. In combination with the median cytokine dose (0.1 ng/µL) with PLX EVs or with HVT EVs, BCL2/BAX expression was increased by 32% (*p* = 0.0571) and by 73% (*p* = 0.0571), respectively, compared to cells exposed to cytokines alone. Only HVT EVs affected the high dose of cytokines (1 ng/µL) and increased the BCL2/BAX expression by 38% (*p* = 0.0571) (Figure 7c). 

#### 2.2.5. Effects of Placental EVs on TGF-β-Induced Fibrosis-Related Proteins in NHDF Cells

We have found that patient EVs contained both significantly higher levels of TGF-β compared to HC EVs and increased TGF-β expression in NHDF cells (Figure 2). We next explored the stimulatory effects of TGF-β with or without placental EVs on the NHDF cell fibrosis pathway. Combining TGF-β and PLX EVs strongly reduced αSMA levels as measured by Cohen’s d analysis (0.796), compared to the effects of TGF-β alone (Figure 8a). Combining TGF-β with HVT EVs reduced the stimulatory effects of TGF-β on the αSMA production by over 40% (*p* = 0.0286) (Figure 8b).

## 3. Discussion

The current study aimed to characterize the EVs of cGVHD patients and validate whether those EVs could serve as potential biomarkers for a cGVHD state. In addition, we aimed to explore the therapeutic potential of placental EVs in dermal inflammatory and fibrotic conditions. The study findings showed that cGVHD patients’ EVs contained cargo with pro-fibrotic characteristics that affected dermal cell function, and that placental EVs can attenuate this cytokine-induced dermal cell inflammation and fibrosis. We also studied the effects of cGVHD patient EVs on dermal cell culture models that could predict the response or lack of response to treatment of cGVHD in an in vitro setting for, what we believe to be, the first time [26]. 

cGVHD is the most common and severe complication in patients surviving more than 100 days after allogeneic HCT. cGVHD involves various immune cells and molecules that affect multiple organs [8]. The expansion of donor T cells inflicts damage to target organs by inducing fibrosis, activating inflammatory cytokines or promoting B-cell activation, which results in the production of autoantibodies [27]. The assessment of disease severity requires the identification of prognostic, diagnostic, and predictive biomarkers of cGVHD [26]. The EV membrane antigens and several microRNAs packed within EVs were suggested as a potential biomarker for assessing cGVHD severity [26]. Given that fibrosis is considered the final stage of the inflammatory response cascade in cGVHD patients, we explored the presence of fibrosis-related proteins in the EVs derived from cGVHD patients compared to those obtained from HC subjects. To the best of our knowledge, our study is the first to propose the cytokine cargo of EVs as a potential reliable biomarker for evaluating the condition of patients with cGVHD. 

TGF-β is a crucial cytokine involved in the activation of fibroblasts into myofibroblast phenotypes that express αSMA, as well as in the production of collagen. In the setting of cGVHD, TGF-β is secreted from mononuclear cells, and it has been recognized as a valuable prognostic marker of cGVHD [28]. The TGF-β signaling pathway is the primary pathway responsible for activating fibrosis-related proteins, and the neutralization of TGF-β has been suggested as a possible therapeutic strategy for preventing cGVHD [29]. SMAD7, on the other hand, is a significant negative regulator of the TGF-β signaling pathway. When TGF-β binds its receptor and triggers downstream signaling, SMAD7 is released from the nucleus and can either inhibit Smad2/3 phosphorylation or induce the degradation of the TGF-β receptor I and Smad2/3 [30]. αSMA plays a significant role in fibrogenesis and in the differentiation of fibroblast to myofibroblasts [31]. Our study revealed that EVs obtained from cGVHD patients exhibited significantly higher levels of TGF-β compared to EVs obtained from HC. This increase in TGF-β correlated positively with the elevated levels of the fibrosis-related proteins, SMAD7 and αSMA. 

In the current study, while 71% of the cGVHD patients were treated with glucocorticoids, known as an inhibitor of TGF-β1 and -β2 production [32], the EVs of 57% of these patients contained higher levels of TGF-β compared to the mean levels that were found in HC EVs, reflecting the treatment resistance. Elevated levels of fibrosis-related proteins in EVs from cGVHD patients may not only serve as biomarkers for disease severity and treatment efficiency, but also contribute to tissue damage and vulnerability in target cells, as occurs in the skin cells of cGVHD patients. 

To assess the involvement of EVs in cGVHD pathology, we studied the effects of patients’ EVs compared to EVs of HC, as well as the effects of TNFα, INFγ, and TGF-β that are known to be involved in the inflammatory and fibrotic stages of cGVHD. The findings of the current study revealed that increasing doses of the cytokines TNFα and INFγ significantly reduced cell proliferation, increased cell apoptosis, and reduced anti-apoptosis-related genes (BCL2/BAX ratio) in stimulated HaCaT cells. Stimulating fibroblast cells with either cGVHD patient EVs or TGF-β increased the fibrotic protein pathway. We consider that a combination of therapeutic EVs originating from placental cells may attenuate these effects.

Mesenchymal stem cells (MSCs) are multipotent, non-hematopoietic stem cells that can be isolated from multiple different tissues, such as the umbilical cord, bone marrow, adipose tissue, and placenta [33]. Human MSCs are used in clinical studies to treat autoimmune diseases, including GVHD. Meta-analyses assessing the use of MSCs for cGVHD patients have suggested that the infusion of MSC can reduce the occurrence of cGVHD [34]. Placental MSCs have stronger immuno-regulatory properties compared to those of umbilical-cord-derived MSCs [10], and PLX platforms have recently been used in phase I and II clinical trials for several indications, including cGVHD [35]. MSCs derived from different placental regions have immunosuppressive characteristics and can be used for tissue repair and immunomodulatory therapy [36]. Finally, MSC therapy is associated with transient fever [37], while MSC EVs are considered to be safer without adverse effects [38]. In the current study, we demonstrated that placental EVs originating from PLX or HVT cells, preserved the MSC characteristics of their parental cells. EVs are cell-derived membranous particles which play important roles in inter-cellular communication and are considered to be highly efficient drug delivery vehicles. Their small size enables them to cross the blood barrier to access injured tissues and penetrate to target cells and deliver their cargo [39]. Additionally, we demonstrated that HVT EVs and PLX EVs were able to bind and penetrate human epidermal keratinocyte and dermal fibroblast cells in vitro, and that they modified the cells’ responses to inflammatory stimulators. Placental cells and their related EVs express HLA-G, which is involved in tuning the immune response [40]. Moreover, a recent study by Stamou et al. demonstrated that inducing the expression of HLA-G on T cells converts them to T-regulatory cells [41] that can modulate immunity and prevent autoimmune disease. The absence of regulatory T cells that control TH1 and TH17 cells results in the autoimmune-mediated pathology of cGVHD. 

In the current study, we found that placental EVs from PLX and trophoblast cells counteracted the inhibitory effects of cytokines (TNFα and INFγ 0.01 ng/µL and 0.1 ng/µL) on cell proliferation in dermal keratinocyte cells. Cell apoptosis increased and anti-apoptotic capabilities decreased at high cytokine doses, while this effect was attenuated when either PLX EVs or HVT EVs were added. Additionally, exposure of dermal keratinocyte cells to high-dose cytokines (TNFα and INFγ 1 ng/µL) increased cell apoptosis and reduced cell anti-apoptotic capabilities, as indicated by the decreased ratio of BCL2 (anti-apoptotic) to BAX (pro-apoptotic) genes compared to non-treated cells. However, when combined with PLX EVs or HVT EVs, these effects were attenuated and the moderate dose reduction effects of cytokines on the anti-apoptotic gene expression and the ratio of BCL2 to BAX were overcome.

This study has some limitations. Its findings are based upon small groups of patients, and samples were obtained from patients with the severe form of cGVHD and not from mild or moderate disease forms, precluding our ability to claim that EVs represented varying disease severity. Due to the limited volume of each sample, samples were examined on most, but not all the assays. Additional studies in larger cohorts with several stages of disease severity are warranted.

## 4. Materials and Methods

### 4.1. Patient Acquisition

The study was conducted between the years 2019 and 2023 at the BMT Unit, Tel Aviv (Sourasky) Medical Center**,** Tel Aviv, Israel. All patients participated in phase 1/2 study of administration PLX-PAD for patients with steroid-refractory cGVHD. The study was approved by the committee (TLV 0766-17; MOH_2018-12-30004869). In addition, thirteen healthy individuals, sex- and age-matched, age ≥ 18 years old, served as a control group registered on clinicaltrials.gov (#NCT04746092, 9 February 2021) [42]. All patients and controls provided informed consent. Patient characteristics are presented in Table 1. 

### 4.2. Patients and Healthy Controls EV Isolation and Characterization

#### 4.2.1. Patients and Healthy Controls EV Isolation 

Patient blood samples were obtained through an intravenous 20–22-gauge cannula. Blood was collected in sodium citrate (1:10) and in EDTA. EVs were isolated as previously described [43,44], according to MISEV2018 [45]. Specifically, platelet-poor plasma (PPP) was obtained after two sequential centrifugations (15 min 1500× *g*, 24 °C) within one hour of collection and frozen in aliquots at −80 °C, as we have done in previous studies [37]. Single freeze–thaw cycles were found not to significantly impact EV size or number [46]. For EV pellets, 150 uL or 500 uL of de-freeze PPPs were centrifuged by MIKRO 220R, rotor 1189-A (Hettich, Tuttlingen, Germany) at 20,000× *g*, 1 h, 4 °C, max acceleration, zero declaration. 

#### 4.2.2. EV Characterization

For transmission electron microscopy (TEM) imaging, the EV pellets were washed with PBS and re-pelleted (1 h, 20,000× *g*, 4 °C). The samples were adsorbed on carbon-coated grids and stained with 2% aqueous uranyl acetate. Samples were examined using a JEM 1400 plus transmission electron microscope (Jeol, Tokyo, Japan).

The size, concentration, and membrane antigen expression of EVs were validated on thawed PPP samples. PPP EV sizes and concentrations were validated by NTA using NS300 Automated Nanoparticle Characterization Instrument (Malvern Instruments LTD, Malvern, UK). The software version build 3.1.54., Camera Type sCMOS was used (Laser Module: NS300, 405 nm). Software settings for analysis were kept constant for all measurements. Capture settings were as follows: camera level = 7, temperature = 25 °C, viscosity = 0.86 cP, and syringe pump speed = 20. Five 30 s videos were recorded per sample in light scatter. Samples were diluted (1:200) with phosphate-buffered saline (PBS) filtered through a 0.1 um membrane. The filtered PBS served as control for each measurement.

EV protein content and exosome markers were analyzed through a Western blot. Thirty microliters of EV pellets obtained from similar PPP volumes (500 µL) or 50 µg of placental EV pellets isolated from cell culture media were combined with x2 lysis buffer (Ray Biotech, Norcross, GA, USA), supplemented with 1% proteinase inhibitor and 1% phosphatase inhibitors (Sigma) containing β-mercaptoethanol (1:20, Bio-Rad, Hercules, CA, USA) [43]. The samples were loaded and separated on 4–20% Mini-PROTEAN TGX Precast Protein Gels (Bio-Rad) and then transferred to Trans-Blot Turbo Mini 0.2μm Nitrocellulose Transfer Packs (Bio-Rad). The transferred membranes were incubated with specific antibodies (Appendix A), documented by a myECL™ Imager and analyzed by the My Image Analysis Software v2.0 (both from Thermo Fisher Scientific, Waltham, MA, USA).

To evaluate placental EV membrane antigens, EVs were assessed by flow cytometry (FACS CANTO, BD Biosciences, San Jose, CA, USA) using Size Mega mix beads (0.5, 0.9, 3 um, Biocytex-7801, Marseille, France) to set the EV size gate. Fluorescent antibodies (Appendix A) were used to characterize mesenchymal phenotypes. Events were collected by time at a flow rate of 10 µL per minute. Controls and samples were analyzed in the same acquisition setting and reagent conditions. 

### 4.3. Cell Culture

#### 4.3.1. Human Early-Stage Trophoblast (HVT) Cells

Primary cells of the second trimester were purchased from Scien-Cell (Cat number 7120, Carlsbad, CA, USA). HVTs were cultured in a complete medium with DMEM/F-12 (HAM) 1:1, 10% heat-inactivated fetal bovine serum (FBS), 1% Pen-strep solution (Penicillin: 10,000 units/mL; streptomycin 10 mg/mL), supplemented with insulin-like growth factor 1 (IGF-1, 15 ng/mL, Peprotech, Rehovot, Israel), and fibroblast growth factors (FGF, 125 pg/mL, Peprotech, Rehovot, Israel). The cells were grown at 37 °C, 5% CO_2_. Passages 6–11 were used for EV isolation. Forty-eight hours before EV isolation, the medium was replaced with a serum-depleted medium, low-glucose DMEM. The reagents for cell biology were obtained from BI Biological Industries, Israel, unless otherwise declared.

#### 4.3.2. Placental-Expanded (PLX) Cells

PLX) cells were manufactured by Pluri Inc. (formerly known as Pluristem Therapeutics inc, Haifa, Israel) from term placentas, as described previously [24]. Thawed cells were grown in a complete medium of DMEM supplemented with 10% heat-inactivated FBS, 1% pen-strep solution (penicillin: 10,000 units/mL; streptomycin 10 mg/mL). Cells were grown for one week. Forty-eight hours before EV isolation, the medium was switched to a serum-free medium, low-glucose DMEM. 

#### 4.3.3. Placental EV Isolation from Cell Medium

For placental EV production, cell media were changed to a medium without serum and EVs were isolated after 24 and 48 h of starvation by several centrifugation steps. First, the cell medium was centrifuged at 400× *g* for 5 min and the cell pellets were discarded. The supernatant was then centrifuged for 15 min at 1500× *g* and the cell debris was subsequently discarded. The supernatant was then submitted to ultracentrifugation (UC) (SORVAL Wx + Ultra series centrifuge, Thermo scientific, rotor: Fiberlite F37 L-8X100. Japan) at 20,000× *g* for 1 h at 4 °C, max acceleration, zero declaration. The supernatant was discarded, and the EV pellets were divided to aliquots of 50 µg and frozen at −80 °C.

The expression of mesenchymal markers on placental cells and their related EVs was validated by fluorescent antibodies (Appendix A) and analyzed by flow cytometry (The BD FACS Canto). For EV analysis, the percentage of labeled EVs was calculated from the total number of EV counts in the EV gate, set by the Mega mix beads (Biocytex), a mix of fluorescent beads at the sizes of 0.5 and 0.9 um.

#### 4.3.4. Primary Human Keratinocytes (HaCaT)

HaCaT cells are the recommended model for investigating anti-inflammatory interventions/therapies for skin diseases [39]. Cells were kindly provided by Dr. Ofer Sarig (Molecular Dermatology laboratory, TASMC, Tel Aviv, Israel). Cells were cultured in 5% CO_2_ at 37 °C in a complete medium, including DMEM supplemented with 10% heat-inactivated FBS, 1% Pen-strep solution (penicillin: 10,000 units/mL; streptomycin 10 mg/mL). 

For all experiments with HaCaT, the cells were seeded at a density of 5000–20,000 cells/well in a 96-well plate. After 24 h, the cell medium was changed to a medium depleted serum. TNFα and INFγ (0.01, 0.1 and 1 ng/µL, Peprotech, Israel), PLX EVs (50 ug/well), HVT EVs (50 µg/well), and a combination of cytokine and placental EVs were added to the cells and compared with non-treated cells. Cell proliferation (after 24 h of stimulation) and cell apoptosis (after 48 h of stimulation) were assessed. The proliferation rate was measured by XTT, a colorimetric assay (BI Biological Industries, Beit Haemek Israel), according to the manufacturing protocol. Apoptotic rates were validated by an annexin/PI assay (MEBCYTO Apoptosis Kit, NBL Life Sciences, Nagoya, Japan) using fluorescent microscopy and flow cytometry analysis. Levels of gene expression were studied after 6 h of cell exposure to EVs, with or without stimulators. Cells were washed with PBS and the total RNA was extracted with a TRI-Reagent kit (Molecular Research Center, Cincinnati, OH, USA). RNA concentration and purity were determined by UV absorption at 260 and 280 nm (NanoDrop). cDNA was constructed with a Maxima First Strand cDNA Synthesis Kit for RT-qPCR (Cat. K1641 Thermo Fisher Scientific). mRNA expressions of BCL2 and BAX were validated by a Taqman gene expression assay and evaluated with a quantitative StepOnePlus real-time PCR system (Applied Biosystems), and compared to the human housekeeping GAPDH gene. All reagents were obtained from Applied Biosystems (Waltham, MA, USA). Results were expressed as a relative quantification. 

#### 4.3.5. Normal Human Dermal Fibroblasts (NHDF) Cells

NHDF cells were purchased from PromoCells and were cultured with DMEM supplemented 10% heat-inactivated FBS, 1% Pen-strep solution (penicillin: 10,000 units/mL; streptomycin 10 mg/mL) and 2 mM glutamine. The cells were maintained in a humidified atmosphere of 95% air and 5% CO_2_ at 37 °C. For all experiments, NHDF cells were seeded at a density of 20,000 cells/well in a 96-well plate. After 24 h, the cell medium was changed to a medium depleted serum. TGF-β (0.25 ng/µL, Peprotech Israel) was added in duplicate, with or without PLX EVs (50 µg/well) or HVT EVs (50 µg/well), and compared with non-treated cells or to cells that were treated with placental EVs only for 48 h. Patient EV vs. HC EV pellets obtained from 120 uL PPP, added in duplicate for 24 h. At the end of each stimulation, the cells were washed and proteins were extracted by buffer lysis x2 (Ray Biotech) for Western blot analysis. Protein concentrations were measured by a BCA Protein Assay Kit (Pierce, IL, USA). WB analysis was performed as described above with EV samples (4.2.2), using specific antibodies for fibrosis-related proteins (Appendix A). 

### 4.4. Placental EV and Cell Interaction

PLX or HVT cells were washed and exposed to calcein AM (2 µM, Invitrogen, Waltham, MA, USA) in a serum-free medium at 37 °C for 2 h, ending with a wash. A new, serum-free medium was added to the cells. After 24 h, EVs were isolated from the cells as previously described. HaCaT cells were seeded on a cover glass (*n* = 7000 cells) for 24 h, then cells were washed and labeled with Hoechst (blue nuclear staining; abcam, Cambridge, UK) and with Vybrant DiD Cell-Labeling Solution (red phospholipid live cell membrane staining; Invitrogen), according to the manufacturer’s instructions, ending with a cell wash. Green PLX EVs and green HVT EVs were added to HaCaT cells on the cover glass for 12 h. The interaction between placenta EVs and HaCaT cells was documented with fluorescent confocal microscopy (Leica SP8 microscope, Wetzlar, Germany). 

HaCaT or NHDF cells (*n* = 7000) were seeded on a 96-well plate for 24 h in complete HaCaT or NHDF medium, and then a new medium free of serum, with 0.34 × 10^10^ particles/mL of green placental EVs, was added to the cells for overnight incubation. The supernatant from PLX or HVT EVs isolation with the green exosome phase was added to the cells as a negative control. Interaction of green placental EVs with non-labeled HaCaT or NHDF cells were analyzed by flow cytometry (FACS CANTO BD).

### 4.5. Statistics

Statistical analysis was performed with the GraphPad Prism 5 software (GraphPad Software Inc., San Diego, CA, USA). Results were assessed by a one-way ANOVA, non-parametric Kruskal–Wallis test, and Dunn’s multiple comparison test. The non-parametric Mann–Whitney U test and Student’s *t*-tests were used when only two groups were compared. A *p*-value < 0.05 was considered statistically significant. Spearman correlations were performed, along with coefficient values (rho) and 95% confidence intervals. Fisher’s exact test was used to determine whether or not there was a significant association between two categorical variables. Effect size analysis was performed using Cohen’s d method to characterize the sizes of the differences between the groups. Small, moderate, and large effects were defined as 0.20, 0.40, and 0.80, respectively [47,48]. *n* = the number of samples that were validated in each subgroup in each figure.

## 5. Conclusions

The EV profile of cGVHD patients differs from that of EVs obtained from healthy individuals, and may serve as a biomarker for disease status. Placental EVs may regulate dermal inflammation and fibrosis. Further research is needed to explore the therapeutic properties of placental EVs.

## Figures and Tables

**Figure 1 ijms-24-08126-f001:**
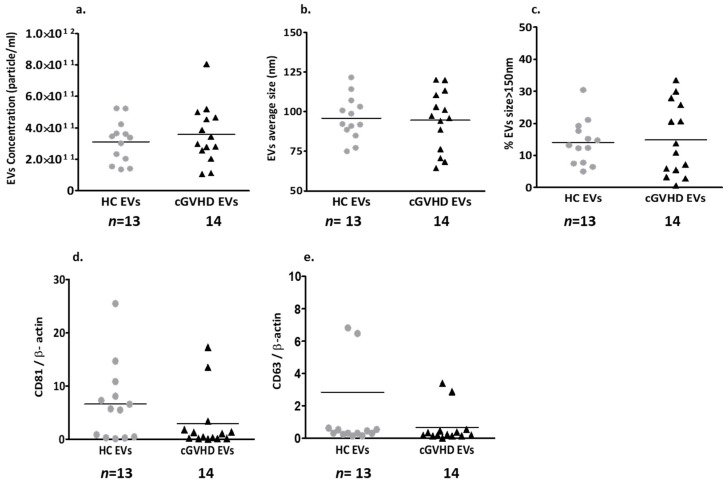
Size, concentration and exosome markers of EVs of patients (triangle) and HC (circle)**.** Extracellular vesicle (EV) content in platelet-poor plasma was measured by nano-tracking analysis (NTA). EV concentration (**a**), size distribution (**b**), and percentage of large EVs (>150 nm) (**c**). Expression of exosome markers, CD63 and CD81, were analyzed through a Western blot and expressed as a ratio of actin (**d**,**e**).

**Figure 2 ijms-24-08126-f002:**
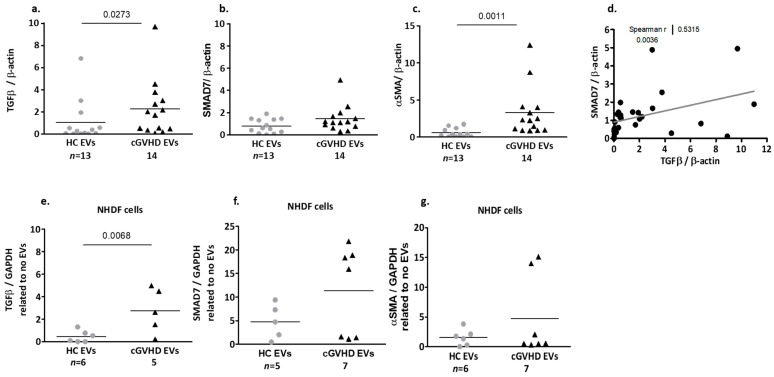
Contents of fibrosis-related proteins in cGVHD patient EVs and HC EVs, and their effects on normal human dermal fibroblast (NHDF) cells**.** Expressions of TGF-β (**a**), SMAD7 (**b**), and αSMA (**c**) in EV pellets were analyzed through a Western blot and expressed as a ratio to actin. Correlation between TGF-β and SMAD7 in HC and cGVHD patient EVs (**d**). Effects of EV pellet isolated from 150 µL PPP of cGVHD patients and HC on NHDF cell fibrosis-related protein expression, expressed as a ratio to GAPDH: TGF-β (**e**) SMAD7 (**f**), and αSMA (**g**).

**Figure 3 ijms-24-08126-f003:**
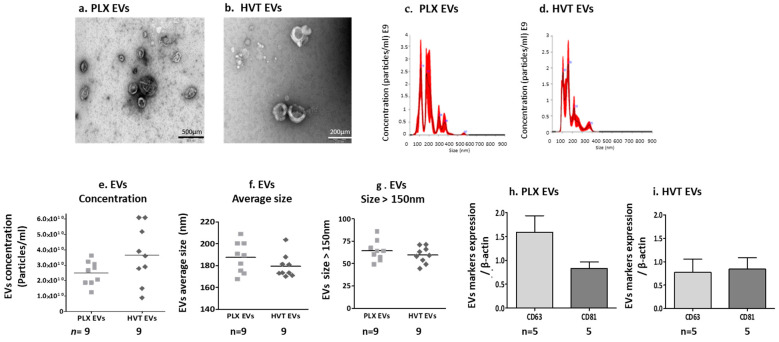
Placental EV size and exosome markers. EVs were isolated from placental cell culture media (PLX and HVT cell) after 48 h of incubation in a serum-free medium. Transmission electron microscopy (TEM) images of EVs (**a**,**b**). EV size distribution graph of a representative sample obtained from PLX (**c**) and HVT (**d**), measured by nano-tracking analysis (NTA). The red shadow indicates error bars (± 1 standard error of the mean of five NTA measurements). EV concentration (particles/mL) (**e**) and EV mean size (**f**). Percent of large EVs (>150 nm) in each sample (**g**). The expression levels of exosome markers CD63, CD81, on PLX EVs (**h**) and HVT EVs (**i**) analyzed through a WB and expressed as a ratio to actin.

**Figure 4 ijms-24-08126-f004:**
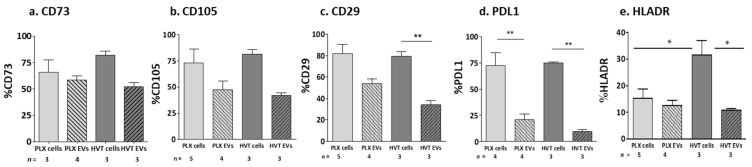
Placental EV MSC markers compared to their parental cells. EVs were isolated from the cell culture media of PLX- and HVT cells. Marker expression were measured via flow cytometry using fluorescent antibodies. CD73 (**a**), CD105 (**b**), CD29 (**c**), PDL1 (**d**), and HLADR (**e**). * *p* < 0.05, ** *p* < 0.01.

**Figure 5 ijms-24-08126-f005:**
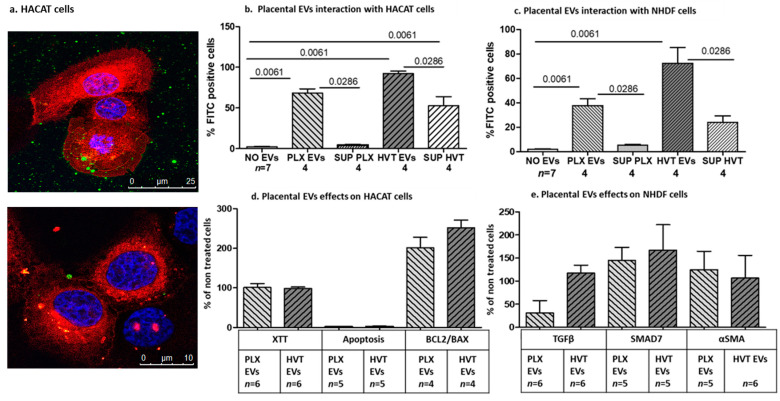
Interaction and effects of placental EVs on dermal cell function. EVs were isolated from calcein-AM-labeled placental cells. The HaCaT cell nuclei were labeled with Hoechst (blue), and the cell membranes were stained with DID (red) for one hour and then washed. PLX EVs (green) were added to HaCaT-labeled cells and documented by confocal microscopy 12 h later. Combinations of green and red are depicted in yellow, combinations of green, red, and blue are depicted in pink (**a**). Green EV pellets isolated from calcein-AM-labeled placental PLX and HVT cells or the supernatant fluid (SUP) were added to non-stained HaCaT (**b**) and NHDF (**c**) dermal cells for 12 h and then washed. The cells were then analyzed with flow cytometry for the percent of FITC-positive cells and compared to non-treated cells. The effects of placental EVs on HaCaT cell proliferation were measured by the XTT method, HaCaT cell apoptosis was measured by annexin/PI-positive cells, and the percentage ratio of BCL2/BAX gene expression was measured by RT-PCR and compared as a ratio of non-treated cells (**d**). Effects of placental EVs on NHDF cell fibrosis-related proteins (TGF-β, SMAD7 and αSMA) were analyzed through a Western blot (WB) and expressed as ratio of non-treated cells (**e**).

**Figure 6 ijms-24-08126-f006:**
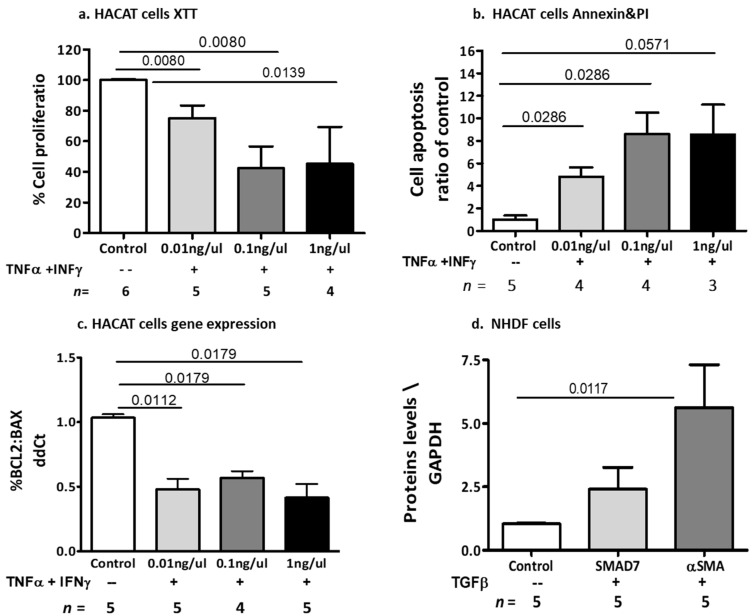
Effects of inflammatory cytokines on dermal cell function. HaCaT cells were exposed to increasing doses of TNFα and IFNγ (0.01 ng/µL, 0.1 ng/µL, 1 ng/µL), and were compared to non-exposed cells. The effects of the cytokines on cell proliferation were measured by an XTT assay after 24 h of treatment (**a**). The effects of the cytokines on cell viability were measured after 48 h of treatment by annexin/PI assay (**b**). The effects of the cytokines on gene expression were measured after 6 h of treatment by the RT-PCR method and expressed as the percentage ratio of BCL2 to BAX expression (**c**). Dermal fibroblast (NHDF) cells were exposed to TGF-β (for 48 h) and the levels of the fibrotic proteins SMAD7 and αSMA were measured through a WB and expressed as a ratio of GAPDH (**d**).

**Figure 7 ijms-24-08126-f007:**
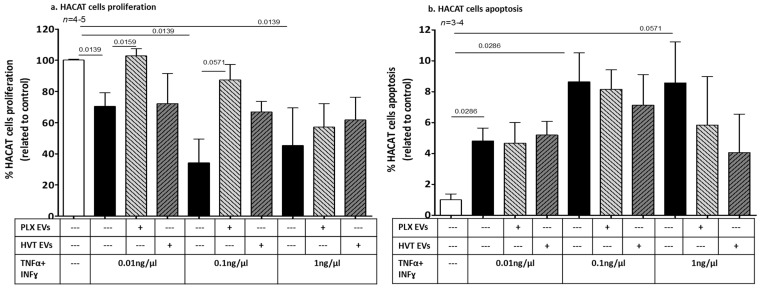
Effects of placental EVs on cytokine-induced keratinocyte cell proliferation and apoptosis. HaCaTcells were seeded on 96 wells at 80% well coverage. Cells were washed and the medium was changed to a medium without serum. Three doses of the TNFα/IFNγ (0.01, 0.1, and 0.1 ng/µL) combination were added with or without PLX EVs (50 µg) or HVT EVs (50 µg), and compared to non-treated cells. Proliferation rates were measured by an XTT assay after 24 h of stimulation (**a**). Apoptosis rates were measured by annexin/PI kit and assessed by flow cytometry after 48 h of stimulation (**b**). The percentage of anti-apoptotic gene expression of BCL2 and BAX were validated by RT-PCR after 6 h of stimulation. The graph presents the ratio between the expressions of the two genes (**c**).

**Figure 8 ijms-24-08126-f008:**
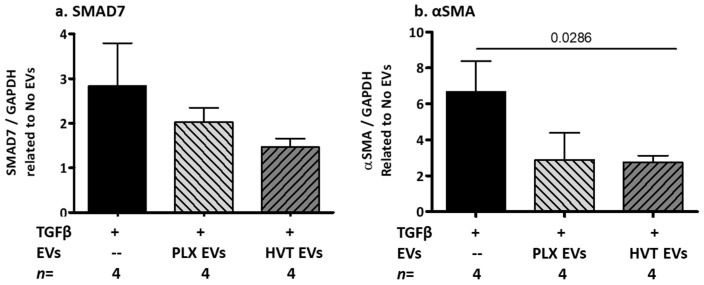
Effects of placental EVs on the fibrosis pathway of TGF-β-stimulated dermal fibroblast (NHDF) cells. NHDF cells were seeded on 96 wells at 80% coverage. The cells were washed and the medium was changed to a medium without serum. TGF-β (0.25 ng/µL) was added with or without PLX EVs (50 µg) or HVT EVs (50 µg) for 48 h and compared to non-treated cells. The cells were lysed in buffer lysis and the levels of SMAD7 (**a**) and αSMA (**b**) were validated by a WB and expressed as ratios of GAPDH.

**Table 1 ijms-24-08126-t001:** Patient clinical characteristics.

Domain	cGVHD Patients (*n* = 14)
Age, median (range)	56 (23–72)
Sex, *n* female (%)	7 (44%)
Ethnicity, *n* (%)	
Ashkenazi Jewish	8 (50%)
Sephardic Jewish	6 (38%)
Arab	2 (12%)
Time from AlloHCT (months), median (range)	
Time from onset of chronic GVHD (months), median (range)	60 (34–134)
Chronic GVHD severity, *n* severe (%)	16 (100%)
Previous lines of treatment, median, range	5 (3–5)
Organs involved, *n* (%)	
Mouth	13 (81%)
Eyes	15 (94%)
Skin	16 (100%)
Fascia	11 (69%)
Joints	9 (56%)
Lungs	11 (69%)
Gastro-intestinal	5 (31%)
Liver	4 (25%)
Medications at the time of plasma sampling	
Prednisone	10 (71%)
Tacrolimus, Cyclosporin	2 (14%)
JAK-STAT inhibitor (Ruxolitinib)	6 (43%)
Tyrosine kinase inhibitor (Imatinib)	2 (14%)
Extra corporal photopheresis	1 (7%)

## Data Availability

Data is unavailable due to privacy and ethical restrictions.

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
