# Peer review of "The Role of Extracellular Vesicles (EVs) in Chronic Graft vs. Host Disease, and the Potential Function of Placental Cell-Derived EVs as a Therapeutic Tool"

_ijms, 2023, doi:10.3390/ijms24098126_

Round 1
Reviewer 1 Report
The paper concerns in vitro effects of extracellular microvesicles (EV) obtained from the patients with chronic GvHD upon human dermal fibroblasts and keratinocytes.
The contents of biomarker molecules (TGFβ and smooth muscle actin) in the patient’s EV was higher than in control EV preparations, and their amounts were increased in normal fibroblasts upon addition of the patients’ EV. EVs from placental and trophoblast cells well penetrated into the model fibroblasts and dermal keratinocytes in vitro. The placental EVs could modulate the in vitro modeled inflammation-like events in epidermal keratinocytes, thus, potentially, preventing fibrosis in GvHD and similar disorders.
Remarks.
Section 4.1 (Patient acquisition). What therapy was applied in the patients with GVHD at the time of plasma sampling (steroids, other immunosuppressors, or none)
Section 4.2.2., line 500: the initial steps of placental tissue incubation should be referred to section 4.3 below.
Section 5 (Conclusions). The authors have shown only a modulatory effect of placenta-derived EVs. Therefore, a suggestion of novel therapeutic approach should be more cautious.
In conclusion, the study is well designed and the results are quite convincing, being in line with current approach in the EV research and deserves publication without gross changes.
Author Response
Answers to Review Report 1
- The manuscript was update based on the reviewer’s comments. A native English-speaking scientific editor checked and edited the new version.
- Corrections were made in the methods section
- The conclusions were edited as suggested by the reviewer.
The paper concerns in vitro effects of extracellular microvesicles (EV) obtained from the patients with chronic GvHD upon human dermal fibroblasts and keratinocytes.
The contents of biomarker molecules (TGFβ and smooth muscle actin) in the patient’s EV was higher than in control EV preparations, and their amounts were increased in normal fibroblasts upon addition of the patients’ EV. EVs from placental and trophoblast cells penetrated into dermal fibroblasts and keratinocytes in vitro. The placental EVs could modulate the in vitro modeled inflammation-like events in epidermal keratinocytes, thus, potentially, preventing fibrosis in GvHD and similar disorders.
Remarks.
Section 4.1 (Patient acquisition). What therapy was applied in the patients with GVHD at the time of plasma sampling (steroids, other immunosuppressors, or none)
Answer
Data regarding cGVHD patient's therapy at the time of plasma sampling was added to Table 1: Patients' clinical characteristics
|
Medications at the time of plasma sampling- n,(%) |
|
|
Prednisone |
10(71%) |
|
Tacrolimus, Cyclosporin |
2(14%) |
|
JAK-STAT inhibitor (Ruxolitinib) |
6(43%) |
|
Tyrosine kinase inhibitor (Imatinib) |
2(14%) |
|
Extra corporal photopheresis |
1(7%) |
|
|
|
In addition, a comment was added in the discussion line 434:
" In the current study, while 71% of the cGVHD patients were treated with Glucocorticoids, known as an inhibitor of TGF-β1 and -β2 production (32), the EVs of 57% of these patients contained higher levels of TGFβ compared to the mean levels that were found in HC EVs, reflecting the treatment resistance. Elevated levels of fibrosis-related proteins in EVs from cGVHD patients may not only serve as biomarkers for disease severity and treatment efficiency…".
Section 4.2.2., line 500: the initial steps of placental tissue incubation should be referred to section 4.3 below.
Answer
Section 4.2.2 was moved to section 4.3 and now appears as 4.3.3 Placental EVs isolation from cells medium
In addition, clarifications regarding times' incubation were added to the results section and to the methods section.
We found a typo error and correction was made in the methods section (4.4) and in figure 5 legend, regarding EVs times' incubation with cells. In both places, it was corrected to 12 hours.
Section 5 (Conclusions). The authors have shown only a modulatory effect of placenta-derived EVs. Therefore, a suggestion of novel therapeutic approach should be more cautious.
Answer
The conclusions were attenuated and edited accordingly, along the manuscript:
- The manuscript title was changed, and "novel" was changed to "potential"
- Abstract conclusion was changed (line 38) "Placental EVs may be used to regulate dermal inflammation and fibrosis, warranting further investigation of their therapeutic potential"
- Section 5 (Conclusions, line 644): " Placental EVs may regulate dermal inflammation and fibrosis. Further research is needed to explore the therapeutic properties of placental EVs."

Reviewer 2 Report
Zavaro M’s text describes the role of extracellular vesicles in cGvHD patients compared to healthy donors and later the EV function obtained from placental derived cells.
In general the text is very interesting because it biologically describes the formation of fibrosis and sclerosis in patients with chronic gvhd.
The information on the text is very numerous and difficult to follow for the reader who does not deal with EV or stem cell transplantation.
The suggestion of this review is to split the work into two parts. the first describing EVs in patients with gvhd and without gvhd and a second part dedicated to the study of general EVs from placental cells.
The abstract section must be rewritten because it is difficult to read.
It is not clear the number of the controls because in the abstract are reported 12 patients and on page 3 13 patients.
Author Response
Answers to Review Report 2
Comments and Suggestions for Authors
Zavaro M’s text describes the role of extracellular vesicles in cGvHD patients compared to healthy donors and later the EV function obtained from placental derived cells.
In general the text is very interesting because it biologically describes the formation of fibrosis and sclerosis in patients with chronic GvHD.
The information on the text is very numerous and difficult to follow for the reader who does not deal with EV or stem cell transplantation.
Answer
Numbers related to patients EVs concentration and size were deleted and the first sentence in 2.1.2 (Patients EV size and exosome markers) was edited accordingly (line 113): “Nanoparticle-tracking analysis (NTA) displayed similar concentrations and sizes of EVs in platelet-poor plasma (PPP) obtained from cGVHD patients and healthy controls (HCs) (Figure 1a,b).”
Numbers related to placental EVs concentration (section 2.2.1) were deleted.
Other numbers were shortened to two digits after the period.
The suggestion of this review is to split the work into two parts. the first describing EVs in patients with gvhd and without gvhd and a second part dedicated to the study of general EVs from placental cells.
Answer
The following sentences were added to the last part of the introduction, line 95: “This is a two-part study. The first part investigates the characteristics of cGVHD patients' EVs compared to those of heathy controls. The second part analyzes EVs derived from placental cells and characterizes their therapeutic properties. Specifically, the current study explored the therapeutic potential of EVs originating from two distinct sources of placental cells: (1) human term placental expanded (PLX) mesenchymal-like adherent stromal cells and (2) early pregnancy stage of human villus trophoblast (HVT) cells that line the placental villi and are in direct content with maternal circulation. We assume that placental EVs express the immunosuppressive antigens characteristic of their parental cell surfaces, and may possibly reduce the inflammation and fibrosis related to cGVHD disease".
Additional changes were done accordingly:
The paragraph on “The effects of Cytokine-induced inflammation and fibrosis on dermal cell models” was moved to section 2.2 “The therapeutic properties placental EVs” as section 2.2.3.
The following sentence was added in the first line of this section (line 277): “In order to validate the ability of placental EVs to control the effects of cytokines on dermal cell models, we needed to first define the effects of the related cytokines, such as TNFα, IFNγ and TGFβ, on dermal inflammation and fibrosis.”
The figure numbers and the supplementary figure numbers were changed according to these changes
The abstract section must be rewritten because it is difficult to read.
Answer
The abstract was edited.
It is not clear the number of the controls because in the abstract are reported 12 patients and on page 3 13 patients.
Answer
Numbers of controls was 13. We had a typo error in the abstract that was corrected.

Round 2
Reviewer 2 Report
The new version of the paper is highly improved. In particular the clarity of exposition is significantly improved.
There are still some typos that can be easily checked and corrected.
No other question. I have no other questions to ask